# Rule-ranking method based on item utility in adaptive rule model



Erna Hikmawati[1], Nur Ulfa Maulidevi[2] and Kridanto Surendro[2]

[1] Doctoral Program of Electrical Engineering and Informatics, School of Electrical Engineering and Informatics, Institut Teknologi Bandung, Bandung, Jawa Barat, Indonesia
[2] School of Electrical Engineering and Informatics, Institut Teknologi Bandung, Bandung, Jawa Barat, Indonesia

## ABSTRACT

**Background:** Decision-making is an important part of most human activities regardless of their daily activities, profession, or political inclination. Some decisions are relatively simple specifically when the consequences are insignificant while others can be very complex and have significant effects. Real-life decision problems generally involve several conflicting points of view (criteria) needed to be considered and this is the reason recent decision-making processes are usually supported by data as indicated by different data mining techniques. Data mining is the process of extracting data to obtain useful information and a promising and widely applied method is association rule mining which has the ability to identify interesting relationships between sets of items in a dataset and predict the associative behavior for new data. However, the number of rules generated in association rules can be very large, thereby making the exploitation process difficult. This means it is necessary to prioritize the selection of more valuable and relevant rules.
**Methods:** Therefore, this study proposes a method to rank rules based on the lift ratio value calculated from the frequency and utility of the item. The three main functions in proposed method are mining of association rules from different databases (in terms of sources, characteristics, and attributes), automatic threshold value determination process, and prioritization of the rules produced.
**Results:** Experiments conducted on six datasets showed that the number of rules generated by the adaptive rule model is higher and sorted from the largest lift ratio value compared to the apriori algorithm.

# INTRODUCTION

Decision-making is an important part of most human activities regardless of daily activities, profession, or political inclination. Some decisions are relatively simple, specifically when the consequences are insignificant while others can be very complex and have significant effects. Real-life decision problems generally involve several conflicting points of view (criteria) which are needed to be considered in making appropriate decisions (*Govindan & Jepsen, 2016*). Recent decision-making processes can be supported with data analysis because decision-makers are required to make the right strategic choices

Corresponding author
Erna Hikmawati,
ernahikma21@students.itb.ac.id

in this current volatile, uncertain, complex, and ambiguous period which is also known as the VUCA period (*Giones, Brem & Berger, 2019*).

Data mining is a method of extracting information and patterns stored in data (*Luna, Fournier-Viger & Ventura, 2019*; *Pan et al., 2017*; *Prajapati, Garg & Chauhan, 2017*; *Ryang & Yun, 2015*; *Selvi & Tamilarasi, 2009*; *Zhang & Zhang, 2002*) and its output can be used to support the decision-making process. It can be applied to both internal and external data due to the possibility of accessing data from anywhere, at any time, and from different sources (*Luna, Fournier-Viger & Ventura, 2019*). The most basic and widely applied concept in data mining is the association rule (*Dahbi, Balouki & Gadi, 2018*; *Duong et al., 2016*; *Lin et al., 2019*; *Luna, Fournier-Viger & Ventura, 2019*; *Prajapati, Garg & Chauhan, 2017*; *Ryang & Yun, 2015*; *Selvi & Tamilarasi, 2009*; *Weng & Chen, 2010*; *Zhang & Zhang, 2002*; *Zhang & Wu, 2011*) which has been discovered to be very important in determining and identifying interesting relationships between sets of items in a dataset and also to predict the association relationships for new data (*Vu & Alaghband, 2014*; *Weng & Chen, 2010*; *Zhang & Wu, 2011*).

The basic concept in association rules is to generate rules based on items occurring frequently in transactions and this normally involves two main processes which include the determination of frequent itemset and the process of forming the rule itself. A frequent itemset is a collection of items occurring more frequently than the threshold value or minimum support specified in the transaction. The association rule looks very simple but has several challenges in its practical application ranging from the usage of very large, multiple, and heterogeneous data sources to the difficulty in the process of determining the minimum support (*Zhang & Wu, 2011*). Moreover, the number of rules generated can be very large, thereby making the exploitation process to be difficult (*El Mazouri, Abounaima & Zenkouar, 2019*). This means it is necessary to prioritize the selection of more valuable and relevant rules to be used in the process (*Choi, Ahn & Kim, 2005*).

Previous studies discussed different methods of prioritizing rules and ELECTRE II was discovered to have the ability of sorting the rules from the best to the worst (*El Mazouri, Abounaima & Zenkouar, 2019*). It is also a multi-criteria decision-making method which is based on the concept of outranking using pairwise comparisons of alternatives related to each criterion. Moreover, ranking results are usually obtained by considering different sizes of association rules with those at the top rank representing the most relevant and interesting (*El Mazouri, Abounaima & Zenkouar, 2019*).

The rules generated from the association rule mining process with several other criteria related to business values are normally presented to the managers involved in the business. *Choi, Ahn & Kim (2005)* proposed a method that can create a synergy between decision analysis techniques and data mining for managers in order to determine the quality and quantity of rules based on the criteria determined by the decision-makers (*Choi, Ahn & Kim, 2005*).

This prioritization association rule method has been used to analyze cases of road accidents which are considered the major public health problems in the world to identify

the main factors contributing to the severity of the accidents. It is important to note that the study was not conducted to optimize transportation safety, but to generate sufficient insight and knowledge needed to enable logistics managers to make informed decisions in order to optimize the processes, avoid dangerous routes, and improve road safety. A large-scale data mining technique known as association rule mining was used to predict future accidents and enable drivers to avoid hazards but it was observed to have generated a very large number of decision rules, thereby making it difficult for the decision-makers to select the most relevant rules. This means a multi-criteria decision analysis approach needs to be integrated for decision-makers affected by the redundancy of extracted rules (*Ait-Mlouk, Gharnati & Agouti, 2017*).

The current rule ranking method, which uses the electree method (*Choi, Ahn & Kim, 2005*; *El Mazouri, Abounaima & Zenkouar, 2019*) and the AHP method (*Choi, Ahn & Kim, 2005*), is a separate process from the rule formation process. So, a separate process is needed to rank the rules that have been generated. And this process is not easy, but it requires the determination of alternatives and criteria from the decision-making team. Determination of alternatives and criteria takes a long time and requires room for discussion. The need for data support and some limitations observed also show the need to transform the association rule more adaptively to user needs. This is indicated by the fact that decision-makers have different criteria to determine the information they need which are needed to be considered in the rule formation process in order to ensure adaptive rules are produced (*Hikmawati, Maulidevi & Surendro, 2020*).

The adaptive rule method proposed by *Hikmawati, Maulidevi & Surendro (2020*, *2021a*, *2021b)* and *Hikmawati & Surendro (2020)* has the ability to determine the frequent itemsets based on the occurrence frequency of an item and another criterion called item utility (*Krishnamoorthy, 2018*; *Lin et al., 2019*; *Liu & Qu, 2012*; *Nguyen et al., 2019*) to produce adaptive rule according to the criteria desired by the user. There is also a function in this model usually used to determine the minimum value of support based on the characteristics of the dataset as well as other criteria added to serve as an assessment tool in the rule formation process which is called adaptive support (*Hikmawati, Maulidevi & Surendro, 2021b*).

The adaptive rule model also has the ability to sort the rules generated using the lift ratio which considers the frequency and utility of the item to ensure the rules produced are sorted from those with the highest lift ratio value which are the most relevant. Therefore, the main contributions of the study are highlighted as follows:

1. The lift ratio was not calculated based only on the frequency of the item but also its utility.
2. The rule ranking method was based on the frequency and utility of the item to ensure the rules produced are sequentially arranged based on the highest lift ratio value.
3. The results of the adaptive rule make the decision-making process easier despite the very large number of rules generated by displaying the rule with the highest lift ratio value first.

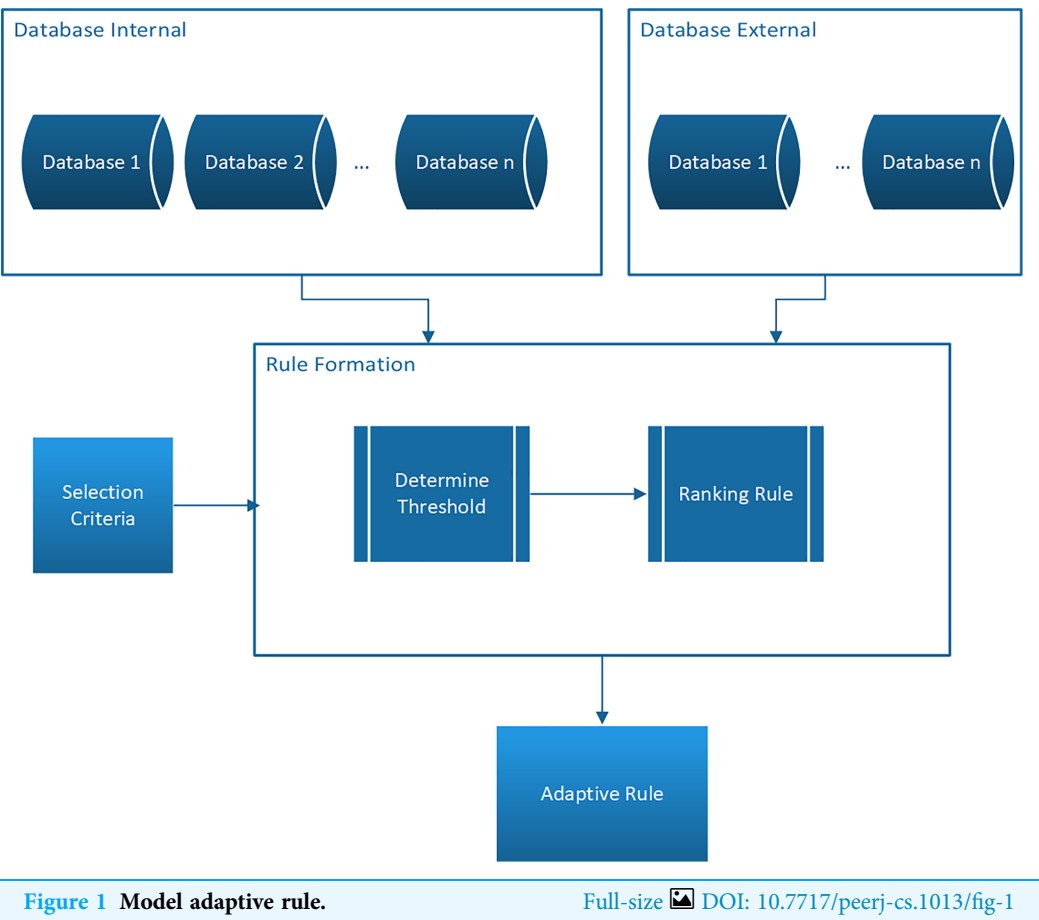

**Figure 1 Model adaptive rule.**

This present study is divided into five parts and the remaining parts are sectioned in such a way that Part II contains related work, Part III explains the proposed method, part IV describes the experimental results, and part V is focused on the conclusions.

## MATERIALS AND METHODS

This present study proposes the adaptive rule model as presented in Fig. 1.

Figure 1 shows that the model has several functions which have been discussed in previous studies (*Hikmawati, Maulidevi & Surendro, 2020, 2021a, 2021b; Hikmawati & Surendro, 2020*) and the three main functions are explained as follows:

1. Mining of association rules from different databases in terms of sources, characteristics, and attributes. It is important to note that the database can either be internal or external (*Hikmawati, Maulidevi & Surendro, 2021a*).

2. Automatic threshold value determination process.

   The system can automatically calculate the threshold value according to the characteristics of the database and the criteria desired by the users and this means there is no need to determine the minimum support and minimum confidence values at the beginning (*Hikmawati, Maulidevi & Surendro, 2021b*).

3. Prioritization of the rules produced

The rules produced are ranked by the lift ratio value based on certain criteria determined by the user. This lift ratio is defined as the ratio between the support value of the rule with the antecedent and consequent support value and can be calculated using the following formula (*Alam et al., 2021*; *El Mazouri, Abounaima & Zenkouar, 2019*; *Kim & Yum, 2011*; *Lin et al., 2019*; *Telikani, Gandomi & Shahbahrami, 2020*; *Tseng & Lin, 2007*):

$$Lift(A \rightarrow B) = \frac{\sup(A \cup B)}{\sup(A)\sup(B)} = \frac{conf(A \rightarrow B)}{\sup(B)} \qquad (1)$$

where:

- Lift is the Lift Ratio Value
- A is the antecedent of the rule in the form of item-set
- B is the consequent of the rule in the form of item-set
- sup is the support value
- conf is the confidence value

There is a slight difference in the adaptive rule model which is associated with the fact that the lift ratio value is not calculated based on only the frequency of the item but also its utility which is a criterion defined by the user. This means the formula to calculate the lift ratio is the same but the support and confidence values generated are not based on only the item frequency. Moreover, the minimum support value is determined automatically using the characteristics of the dataset and the criteria specified by the user. The algorithm in the adaptive rule model is, therefore, presented in Algorithm 1 while the complete stages of the model are indicated in Fig. 2.

Based on the flow of the adaptive model in Fig. 2, the adaptive rule algorithm can be explained as follows:

1. This algorithm starts by preparing input data. There are two types of input obtainable from several different databases in the model and these include:

a. Transaction Dataset: This dataset contains a list of transactions and items in each transaction and is normally used as the basis to form the rule. In the case of sales, this dataset is a collection of sales transactions containing items purchased in the same basket or a collection of purchase receipts. Some of its attributes include the transaction ID and the items purchased for each transaction.

b. Specific criteria (utility for each item)

This utility data can be obtained from external or internal databases and normally applied as the factor to determine the frequent itemset. This means it is possible to have items that rarely appear in the transactions but possess high utility value in the rule formation process. In real cases, the utility can be determined from the price, profit, customer reviews, and availability of goods. It is also important to note that each item can have a different utility value and this makes the rule formation process to be more

**Algorithm 1  Adaptive Rule algorithm**

*Pseudocode adaptive_rule*

*Declare*

> *D = Transaction Dataset*
>
> *S = support value for each item*
>
> *X = the criterion value of an item*
>
> *U = Item Utility value for each item*
>
> *n = frequency of occurrence of an item*
>
> *|D| = number of items in the dataset*
>
> *I = Itemset*
>
> *Function getminsup()*
>
> *Function getFreqItemSet()*
>
> *Function subsets()*
>
> *Function joinSet()*
>
> *Function getSupport()*

*Begin*

> *Function getminsup(itemset, transactionList):*
>
> > *For item in itemset:*
> >
> > > *For transaction as transactionList:*
> > >
> > > > *If item.issubset(transaction) then*
> > > >
> > > > > *freqSet[i]+ = 1*
> > > >
> > > > *End If*
> > >
> > > *End For*
> >
> > *End For*
> >
> > *Sum = 0*
> >
> > *For item, count in freqSet.items():*
> >
> > > *Support = count / len(transactionList)*
> > >
> > > *ut = utilSet[item]*
> > >
> > > *Utility = support\*ut*
> > >
> > > *Sum = Sum+Utility*
> >
> > *Aveutil = Sum/len(itemset)*
> >
> > *Minsup = Avesup/len(transactionList)*

*Return minsup*

*Function getFreqItemSet(itemset, transactionList, minsup, freqSet,k):*

> *_itemSet = set()*
>
> *localSet = defaultdict(int)*
>
> *For item in itemset:*
>
> > *For transaction in transactionList:*

**Algorithm 1** (continued)

> *If item.issubset(transaction) then*
>> *Freqset[item]+ = 1*
>> *localSet[item]+ = 1*
>
>> *End If*
>
>> *End For*
>
> *End For*
>
> *sum = 0*
>
> *for item, count in localSet.items():*
>> *support = count/len(transactionList)*
>> *ut = 0*
>> *for item1,util in utilSet.items():*
>>> *if item1.issubset(item) then*
>>> *ut = ut+util*
>>> *averageutil = ut/k*
>>> *utility1 = support\*averageutil*
>>> *utilSet2[item] = utility1*
>>
>>> *end if*
>>
>> *end for*
>
> *end for*
>
> *for item in itemset:*
>> *if(utilSet2[item]> = minsup) then*
>>> *\_itemSet.add(item)*
>>
>> *End if*
>
> *End for*

*Return \_itemSet*

*Function subsets(arr):*
> *return chain(\*[combinations(arr, i + 1) for i, a in enumerate(arr)])*

*Function joinSet(itemset, length):*
> *itemSet = frozenset(itemSet)*
>> *return set(*
>>> *[i.union(j) for i in itemSet for j in itemSet if len(i.union(j)) = = length]*
>>
>> *)*

*Function getSupport(item):*
> *Return utilSet2[item]*

*minsup = getminsup(I,D)*

*oneCSet = getfreqitem2(I,D, minsup,freqSet,1)*

*currentLSet = oneCSet*

(Continued)

**Algorithm 1 (continued)**

```
k = 2
while currentLSet ! = set([]):
    largeSet[k - 1] = currentLSet
    currentLSet = joinSet(currentLSet, k)
    currentCSet = getfreqitem2(
        currentLSet, transactionList, minsup,freqSet,k
    )
    currentLSet = currentCSet
    k = k + 1
end while
toRetItems = []
for key, value in largeSet.items():
    toRetItems.extend([(tuple(item), getSupport(item)) for item in value])
end for
toRetRules = []
for key, value in list(largeSet.items())[1:]:
    for item in value:
        _subsets = map(frozenset, [x for x in subsets(item)])
        for element in _subsets:
            remain = item.difference(element)
            if len(remain) > 0:
                confidence = getSupport(item) / getSupport(element)
                if confidence >= minConfidence:
                lift=confidence/getSupport(remain)
                    toRetRules.append(((tuple(element), tuple(remain)), confidence,lift))
                end if
            end if
        end for
    end for
end for
from operator import itemgetter
res = sorted(toRetRules, key = itemgetter(2),reverse=True)
print("\n——————————————————————— RULES:")
for rule, confidence,lift in (res):
    pre, post = rule
    print("Rule: %s ==> %s, %.3f, %.3f" % (str(pre), str(post), confidence, lift))
end for
End
```

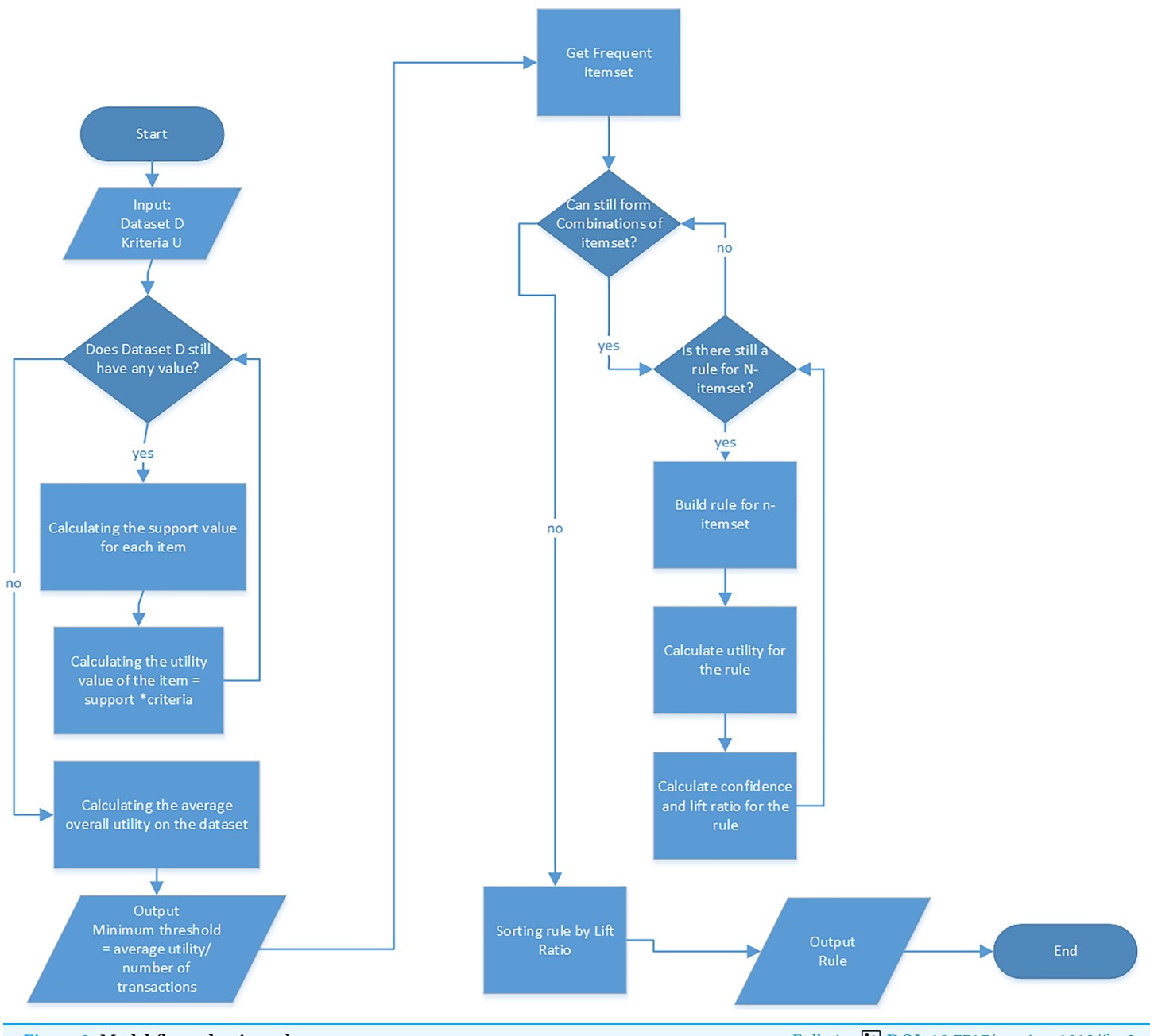

**Figure 2  Model flow adaptive rule.**

adaptive to the needs of the users. Moreover, the user can determine the utilities to be considered in the rule formation process apart from the occurrence frequency of items.

2. From the two types of inputs, an iteration is carried out for the process of calculating the minimum threshold value with the aim of determining the frequent itemset. So that in the adaptive rule model the user does not need to determine the minimum support value at the beginning. The iteration is done as many as the number of items in the dataset. The minimum threshold calculation process follows the following steps:

a. Calculating the support value for each item in the dataset using the following formula:

$$support = count/len(transactionList) \tag{2}$$

b. Calculating the utility for each item in the dataset using the following formula:

$$Utility = support * ut \tag{3}$$

3. If all the items have calculated their utility values, then the iteration process is stopped. The next step is to calculate the average utility for all transactions with the following formula:

$$aveutil = sum/len(itemset) \tag{4}$$

4. The output for this stage is the minimum threshold value used for the rule formation process. The minimum threshold value is obtained from the average utility value divided by the total existing transactions, and this is represented mathematically as follows:

$$minsup = avesup/len(transactionList) \tag{5}$$

where:

   support = support value for an item
   count = number of occurrences for an item
   len(transactionList) = transaction amount
   ut = utility value for an item
   Utility = utility and support value for an item
   Aveutil = Average utility of items
   Sum = sum of utility all item
   Len(itemset) = number of items
   Minsup = minimum threshold value (item density level)

5. After obtaining the minimum threshold value, the items that will be involved in the rule formation process are determined which are called frequent itemset. Items that have utility more than equal to the minimum threshold value are included in the frequent itemset.

6. The next process is the formation of rules in the adaptive rule model based on the apriori algorithm and this was conducted through the following steps:

a) The process in the apriori algorithm is in the form of iterations to form a combination of n-itemsets, starting with a two-itemset combination. The loop is stopped when the item can no longer be combined.

b) The process of forming rules for the combination

c) Calculation of the utility value for the combination such that when the utility value is below the minimum threshold, the rule is eliminated

d) Calculation of the confidence value for each rule such that when the confidence value is below the minimum confidence, the rule is eliminated

**Table 1  Transaction list.**

| Transaction ID | Item 1 | Item 2 | Item 3 |
|---|---|---|---|
| 1 | 1 | 2 | 3 |
| 2 | 1 | 3 | 4 |
| 3 | 1 | 2 | 5 |
| 4 | 1 | 3 | 5 |

**Table 2  Utility items.**

| Item | Utility |
|---|---|
| 1 | 2 |
| 2 | 5 |
| 3 | 3 |
| 4 | 5 |
| 5 | 0.5 |

**Table 3  Calculate utility items.**

| Item | Support | Criteria | Utility |
|---|---|---|---|
| 1 | 1.00 | 2 | 2 |
| 2 | 0.50 | 5 | 2.5 |
| 3 | 0.75 | 3 | 2.25 |
| 4 | 0.25 | 5 | 1.25 |
| 5 | 0.50 | 0.5 | 0.25 |

e) If more itemset combinations can be made, then repeat from Step a.

7. The next step after rule formation is sorting the rules based on the lift ratio value from the biggest to the smallest which was determined using both the frequency and utility of the item. It is, however, important to note that the support value is in the average of the utility when the antecedent or consequent consists of several items.

8. The output of this model is a set of rules that have been sorted based on the lift ratio value.

An example of the implementation of the proposed method can be explained as follows:

1. Input data in the form of a transaction dataset consisting of four transactions can be seen in Table 1.

2. The second input is in the form of utility data items seen in Table 2.

3. The next process is to calculate the minimum threshold by calculating the support for each item and multiplying by its utility. The results can be seen in Table 3.

4. From the results of calculating the utility of each item, the average utility value and the minimum threshold value are calculated.

**Table 4 Two-itemset combination.**

| Item 1 | Item 2 | Frekuensi | Support | Average criteria | Utility | Confidence | Lift |
|---|---|---|---|---|---|---|---|
| 1 | 2 | 2 | 0.5 | 3.5 | 1.75 | 0.875 | 0.35 |
| 1 | 3 | 2 | 0.5 | 2.5 | 1.25 | 0.625 | 0.277778 |
| 1 | 4 | 1 | 0.25 | 3.5 | 0.875 | 0.4375 | 0.35 |
| 2 | 1 | 2 | 0.5 | 3.5 | 1.75 | 0.7 | 0.35 |
| 2 | 3 | 1 | 0.25 | 4 | 1 | 0.4 | 0.177778 |
| 2 | 4 | 0 | 0 | 5 | 0 | 0 | 0 |
| 3 | 1 | 3 | 0.75 | 2.5 | 1.875 | 0.833333333 | 0.416667 |
| 3 | 2 | 1 | 0.25 | 4 | 1 | 0.444444444 | 0.177778 |
| 3 | 4 | 1 | 0.25 | 4 | 1 | 0.444444444 | 0.355556 |
| 4 | 1 | 1 | 0.25 | 3.5 | 0.875 | 0.7 | 0.35 |
| 4 | 2 | 0 | 0 | 5 | 0 | 0 | 0 |
| 4 | 3 | 1 | 0.25 | 4 | 1 | 0.8 | 0.355556 |

Aveutil = 1.65

Min threshold = 0.4125

5. The frequent itemset is 1,2,3,4 because the utility value is >0.4125

6. From the frequent itemset, a two-itemset combination is drawn up which can be seen in Table 4. In addition, the confidence value and lift ratio value are calculated for each rule. Rules that have a utility value <0.4125 will be eliminated and not included in the next itemset combination process.

7. In Table 4 it can be seen that for items 2→4 and 4→2 do not meet the min threshold value so that a three-itemset combination is prepared which can be seen in Table 5.

8. There are several rules that have a utility value <0.4125 so they are not included in the list of rules. And from the existing results, it is not possible to arrange a four-itemset combination.

9. The existing rules are sorted by the value of the lift ratio so that the final result can be seen in Table 6.

The dataset used for this experiment, as previously described in Erna (*Hikmawati, Maulidevi & Surendro, 2021b*), is a special dataset for the rule association case. This dataset is obtained from SPMF (*Fournier-Viger et al., 2016*), UCI Dataset (*Casey & Dua, 2019*) and real transaction data. SPMF is an open-source data mining library in which there are 224 mining algorithms. In addition, various data sets are provided that can be used for data mining. Description of the dataset can be seen in Table 7 and the characteristics of dataset can be seen in Table 8.

# RESULTS

The experimental instrument used was a laptop with an Intel Core i7-8550U CPU @ 1.80 GHz 1.99 GHz, 16 GB Installed memory (RAM), and a 500 GB SSD Hard drive. The proposed adaptive rule model was applied to six datasets according to Table 1. It is

**Table 5 Three-itemset combination.**

| Item 1 | Item 2 | Item 3 | Frequency | Support | Average criteria | Utility | Confidence | Lift |
|---|---|---|---|---|---|---|---|---|
| 1 | 2 | 3 | 1 | 0.25 | 3.3 | 0.825 | 0.4714286 | 0.209524 |
| 1 | 2 | 4 | 0 | 0 | 4 | 0 | | |
| 1 | 4 | 2 | 0 | 0 | 4 | 0 | | |
| 1 | 4 | 3 | 1 | 0.25 | 3.3 | 0.825 | 0.9428571 | 0.419048 |
| 2 | 1 | 3 | 1 | 0.25 | 3.3 | 0.825 | 0.4714286 | 0.209524 |
| 2 | 1 | 4 | 0 | 0 | 4 | 0 | | |
| 2 | 3 | 1 | 1 | 0.25 | 3.3 | 0.825 | 0.825 | 0.4125 |
| 2 | 3 | 4 | 0 | 0 | 4 | 0 | | |
| 3 | 1 | 2 | 1 | 0.25 | 3.3 | 0.825 | 0.44 | 0.176 |
| 3 | 1 | 4 | 1 | 0.25 | 3.3 | 0.825 | 0.44 | 0.352 |
| 3 | 2 | 1 | 1 | 0.25 | 3.3 | 0.825 | 0.825 | 0.4125 |
| 3 | 2 | 4 | 0 | 0 | 4.3 | 0 | | |
| 3 | 4 | 1 | 1 | 0.25 | 3.3 | 0.825 | 0.825 | 0.4125 |
| 3 | 4 | 2 | 0 | 0 | 4.3 | 0 | | |
| 4 | 1 | 2 | 0 | 0 | 4 | 0 | | |
| 4 | 1 | 3 | 1 | 0.25 | 3.3 | 0.825 | 0.9428571 | 0.419048 |
| 4 | 3 | 1 | 1 | 0.25 | 3.3 | 0.825 | 0.825 | 0.4125 |
| 4 | 3 | 2 | 0 | 0 | 4.3 | 0 | | |

**Table 6 Result rules.**

| Ranking | Rule | | Lift |
|---|---|---|---|
| | Anteseden | Konsekuen | |
| 1 | 1,4 | 3 | 0.419048 |
| 2 | 4,1 | 3 | 0.419048 |
| 3 | 3 | 1 | 0.416667 |
| 4 | 2,3 | 1 | 0.4125 |
| 5 | 3,2 | 1 | 0.4125 |
| 6 | 3,4 | 1 | 0.4125 |
| 7 | 4,3 | 1 | 0.4125 |
| 8 | 4 | 3 | 0.355556 |
| 9 | 3 | 4 | 0.355556 |
| 10 | 3,1 | 4 | 0.352 |
| 11 | 1 | 2 | 0.35 |
| 12 | 1 | 4 | 0.35 |
| 13 | 2 | 1 | 0.35 |
| 14 | 4 | 1 | 0.35 |
| 15 | 1 | 3 | 0.277778 |
| 16 | 1,2 | 3 | 0.209524 |
| 17 | 2,1 | 3 | 0.209524 |
| 18 | 2 | 3 | 0.177778 |
| 19 | 3 | 2 | 0.177778 |
| 20 | 3,1 | 2 | 0.176 |

**Table 7 Description dataset.**

| No | Dataset | Description |
|----|---------|-------------|
| 1 | Zoo | A simple database containing 17 Boolean-valued attributes. The "type" attribute appears to be the class attribute. |
| 2 | Connect | This database contains all legal 8-ply connect-4 positions in which neither player has yet won and the following move is not forced. |
| 3 | Retail | Retail Transaction Data from a Store in Nganjuk, East Java |
| 4 | Test | Example transaction dataset |
| 5 | Foodmart | Dataset of customer transactions from a retail store |
| 6 | Sco | Transaction data from a cafe in Bandung, West Java |

**Table 8 Characteristics of the dataset.**

| Dataset | Number of transactions | Number of items | Average number of items on each transaction | Data source |
|---------|------------------------|-----------------|---------------------------------------------|-------------|
| Zoo | 101 | 8 | 17 | https://archive.ics.uci.edu/ml/datasets/Zoo |
| Connect | 100 | 130 | 11 | https://www.philippe-fournier-viger.com/spmf/datasets/quantitative/connect.txt |
| Retail | 72 | 100 | 3 | Retail Transaction Data from a Store in Nganjuk, East Java |
| Test | 8 | 6 | 3 | Test Data |
| Foodmart | 100 | 216 | 3.59 | https://www.philippe-fournier-viger.com/spmf/datasets/quantitative/foodmart.txt |
| Sco | 51 | 22 | 2.52 | Transaction data from a cafe in Bandung, West Java |

**Table 9 Results of formation of adaptive rule.**

| Nama dataset | Adaptive support | Jumlah frequent itemset | Jumlah rule | Runtime (second) | Memory (bytes) |
|--------------|------------------|-------------------------|-------------|------------------|----------------|
| Zoo | 0.006932 | 26 | 29 | 0.02099180 | 29,556,736 |
| Connect | 2.29782 | 21,692 | 950,249 | 72.10392951 | 571,183,104 |
| Retail | 0.001388 | 8,647 | 1,572,003 | 136.73831272 | 743,510,016 |
| Test | 0.20833 | 25 | 43 | 0.01395416 | 36,106,240 |
| Foodmart | 0.214029 | 2,536 | 37,604 | 1.02857613 | 75,522,048 |
| Sco | 0.04260599 | 152 | 258 | 0.04630136 | 37,416,960 |

important to note that price item was another criterion used to determine the minimum threshold in this experiment and the results of the adaptive rule formation process are presented in Table 9.

The results of the adaptive rule test were compared through a basic association rule model trial which involved using the apriori algorithm with the same six datasets. The process was conducted with a level-wise approach such that candidate items were generated for each level (*Agrawal, 1994*) and the same minimum support values were used. The findings of this apriori algorithm method are presented in Table 10.

The comparison of the number of frequent itemsets for the adaptive rule method and the apriori algorithm can be seen in Figs. 3 and 4.

**Table 10  Results of rule formation with the apriori algorithm.**

| Nama dataset | Minimum support | Jumlah frequent itemset | Jumlah rule | Runtime (second) | Memory (bytes) |
|---|---|---|---|---|---|
| Zoo | 0.006932 | 55 | 85 | 0.02099180 | 29,556,736 |
| Connect | 2.29782 | 0 | 0 | 0.02799129 | 14,843,904 |
| Retail | 0.001388 | 8,652 | 1,568,105 | 125.96720266 | 57,773,260 |
| Test | 0.20833 | 16 | 13 | 0.01177000 | 14,893,056 |
| Foodmart | 0.214029 | 0 | 0 | 0.01296401 | 14,802,944 |
| Sco | 0.04260599 | 23 | 1 | 0.01589536 | 14,880,768 |

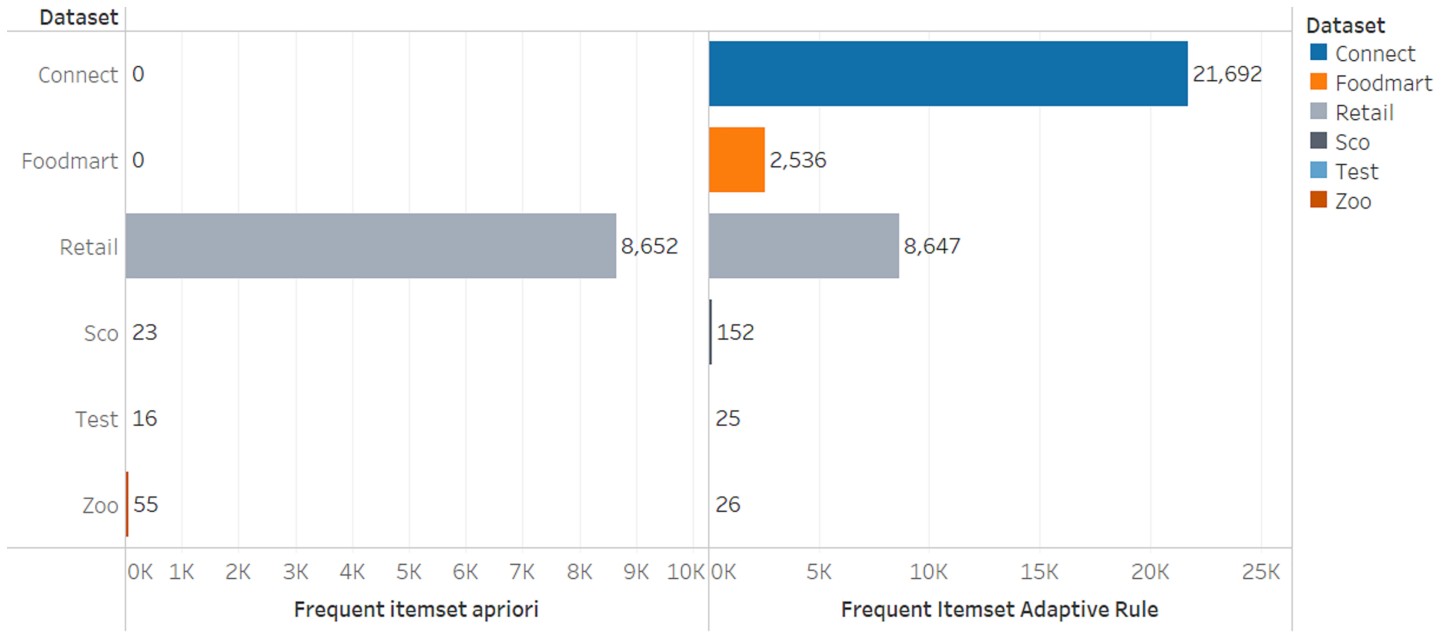

**Figure 3  Number of frequent itemsets.**

The comparison of the runtime and memory consumption for the adaptive rule method and the apriori algorithm can be seen in Figs. 5–7.

In addition, we have conducted an experimental design with a t-test between the number of rules, the number of frequent itemset, runtime and memory. The results of the t-test can be seen in Table 11.

When viewed from the $P$-value > alpha, the alpha here is 0.05, which means that there is no significant difference between the adaptive rule method and the apriori algorithm. In terms of the number of rules, the number of frequent itemset, runtime and memory, there is no significant difference, but this method aims to produce the most relevant rules at the top. So it cannot be measured from the number of rules, the number of frequent itemset, runtime and memory. One of the limitations of the current adaptive rule method is that it has only been tested on a dataset with a relatively small number of transactions, namely 100 transactions, for further research, pruning and performance

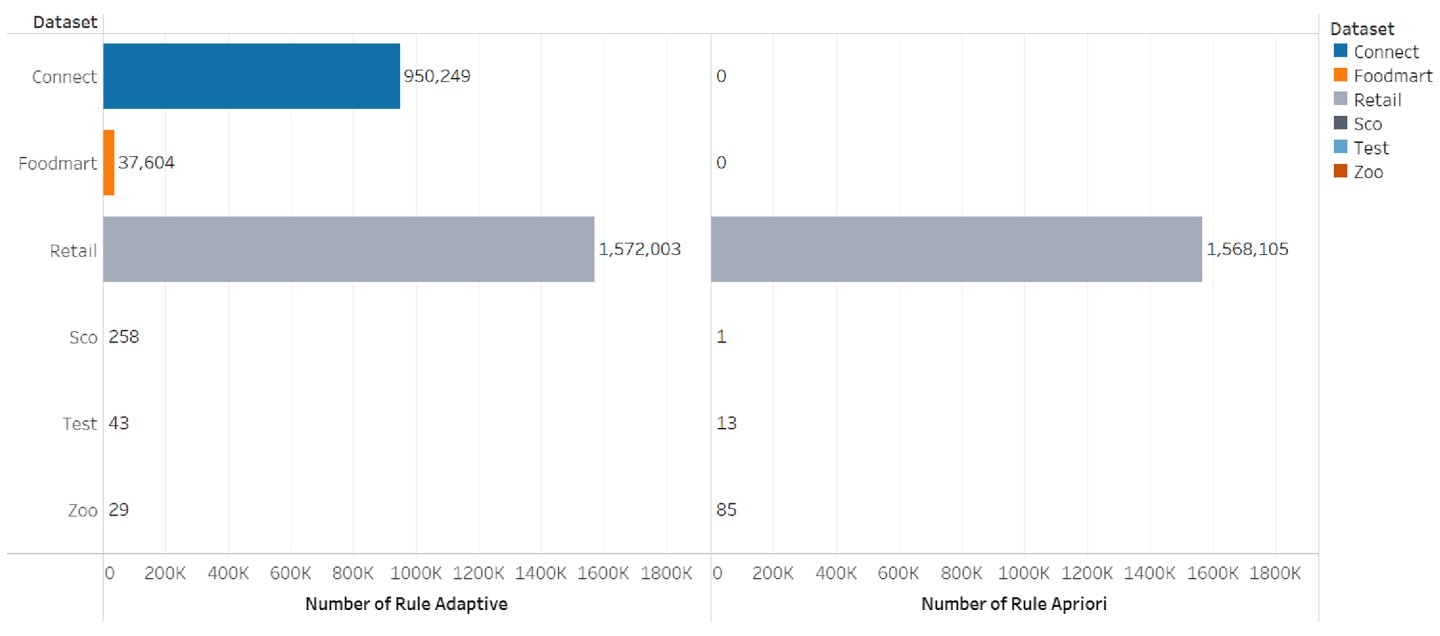

**Figure 4  Number of rules.**               

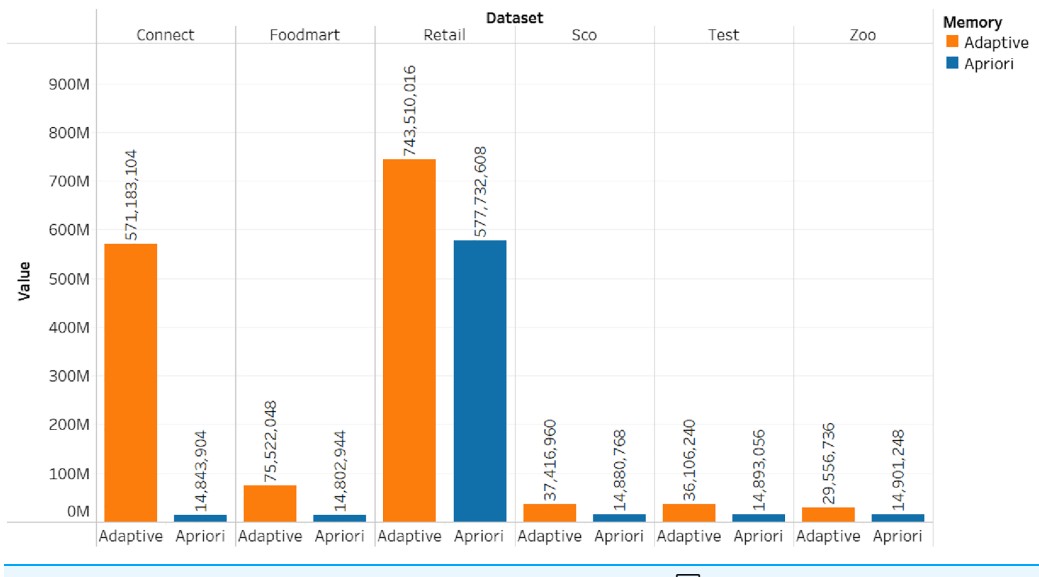

**Figure 5  Memory consumption.**            

improvement will be carried out so that it can be implemented on large datasets. And then a suitable evaluation method will be carried out to measure the success of this adaptive rule method.

## DISCUSSION

Tables 9 and 10 show that the number of frequent itemset and rules generated by the adaptive rule is more than those produced by the apriori algorithm and this means it

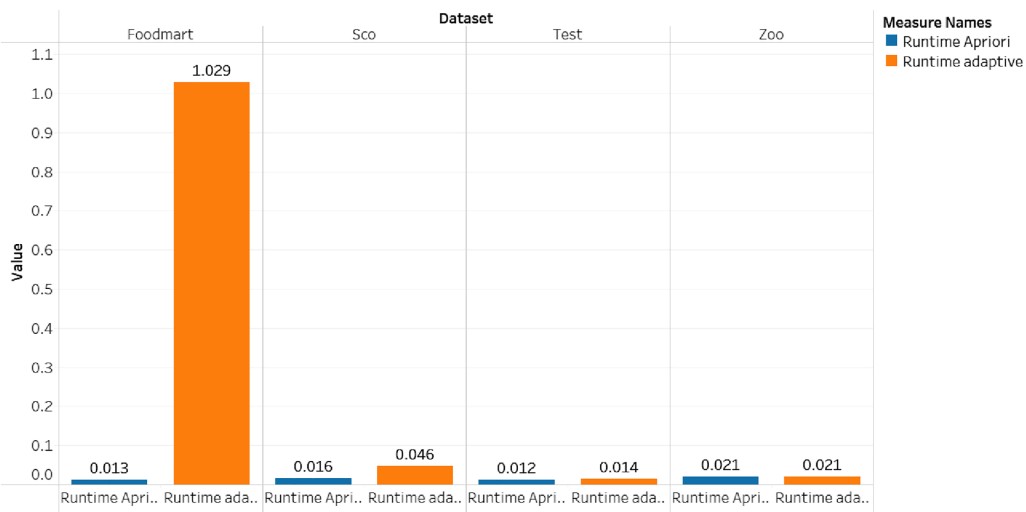

**Figure 6** Runtime for dataset foodmart, sco, test and zoo.

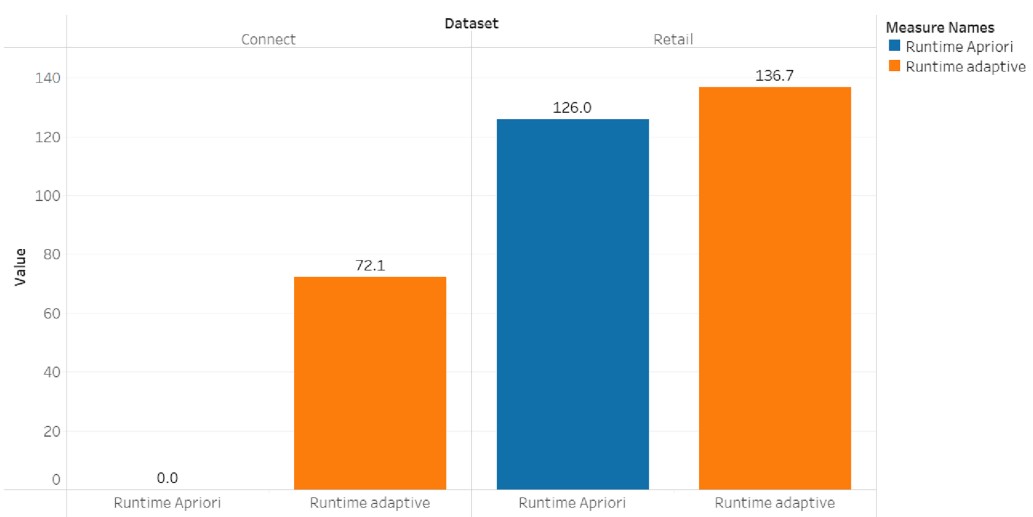

**Figure 7** Runtime for dataset connect and retail.

**Table 11 Result from t-test.**

| Variable | P-value |
| --- | --- |
| Number of rules | 0.341 |
| Number of frequent itemset | 0.305 |
| Runtime | 0.287 |
| Memory | 0.166 |

provides more recommendation options for users. Moreover, the model allows the users to select appropriate choices despite the higher number of options due to its ability to sort the rules automatically based on the lift ratio value.

The number of frequent itemset generated by the adaptive rule is more because the determination of the frequent itemset is not only based on frequency but also involves the utility of the item in this case the price of the item. This allows items that rarely appear in transactions but have a high utility value to be categorized as frequent itemset so that they can be involved in the rule formation process. With the selection of a more varied frequent itemset, it also causes the number of rules generated to be more and can provide more diverse choices for users according to the specified utility criteria. If this is implemented in the recommender system, it will create recommended items not only items that often appear in transactions but which have high utility so that they can provide higher value to users.

Several previous studies focused on ranking rules and some are stated as follows:

1. *El Mazouri, Abounaima & Zenkouar (2019)* used the ELECTRE II method (*Govindan & Jepsen, 2016*) to sort the rules produced from the apriori algorithm while the multicriteria decision analysis method was applied to identify the most frequent conditions in accidents reported in France. The ELECTRE II method was applied to sort a large number of association rules generated (*El Mazouri, Abounaima & Zenkouar, 2019*) from best to worst based on their sizes with those ranked top observed to be representing the most relevant and interesting association rules.

2. *Choi, Ahn & Kim (2005)* sorted the rules generated in the association rule mining process using the ELECTRE and AHP methods. The study focused on prioritizing the association rules produced from data mining through the explicit inclusion of conflicting business value criteria as well as the preferences of the managers concerning trade-off conditions. The decision analysis method such as the Analytic Hierarchy Process (AHP) was used to collect the opinion of group decision-makers on relevant criteria to evaluate the business value of the rules and the relative importance of these criteria. Meanwhile, the association rule mining technique was later used to capture the competing set of rules with different business values which were further applied as input for rule prioritization. This means the final rule was selected from the appropriate decision method such as ELECTRE which was able to present meaningful results using machine learning and human intelligence.

3. *Ait-Mlouk, Gharnati & Agouti (2017)* also conducted a study to generate sufficient insight and knowledge to allow logistics managers to make informed decisions towards optimizing the processes, avoiding hazardous routes, and improving road safety. A large-scale data mining technique known as association rule mining was used to predict future accidents and enable drivers to avoid hazards but it was observed to have generated a very large number of decision rules, thereby making it difficult for the decision-makers to select the most relevant rules. This means a multi-criteria decision analysis approach needs to be integrated for decision-makers affected by the redundancy of extracted rules.

**Table 12 Comparison between previous method and proposed method.**

| No | Title | Method/ Algorithm | Strength | Weakness |
|---|---|---|---|---|
| 1 | Data mining combined to the multicriteria decision analysis for the improvement of road safety: case of France | ELECTREE II | The integration of multi-criteria decision analysis can provide a solution for selecting the most relevant and interesting rules | Using a basic apriori algorithm that must determine the minimum support value and the rule ranking process is a separate process so it takes a long time. |
| 2 | Prioritization of association rules in data mining: Multiple criteria decision approach | ELECTREE and AHP | Can create a synergy between decision analysis techniques with data mining that can be used by managers in determining the quality and quantity of rules that have been determined by decision makers. | Using traditional association rule mining, which requires determining the minimum support value at the beginning. In addition, decision makers need to determine criteria and alternative comparisons to give weight to each rule. |
| 3 | An improved approach for association rule mining using a multi-criteria decision support system: a case study in road safety | Multi-criteria decision support system | Can predict future accidents and allow drivers to avoid hazards. | Generates a lot of rules and there is a lot of redundancy. |
| 4 | Adaptive Rule Model | Automation Minimum Threshold and Rule prioritization by Lift Ratio | Determine the minimum threshold value automatically based on the characteristics of the dataset and certain criteria. The resulting rule has been sorted based on the lift ratio which is calculated not only from the frequency but also by the utility of the item. | It needs pruning method and performance improvement if applied to large amount of data. |

When compared with previous research, the adaptive rule model has the advantage of ranking rules based on the lift ratio which is calculated not only from frequency but also involves utility items such as price. In addition, the rule ranking process does not become a separate part so that when the rules are formed they are automatically sorted.

The comparison between the previous method and the proposed method can be seen in Table 12.

## CONCLUSIONS

The experiment conducted on the adaptive rule model using six datasets was used to draw the following conclusions:

1. The number of rules generated in the six datasets was higher when compared to the apriori algorithm with the same minimum support value.
2. The frequent itemset selection process was not based only on frequency but also considered the specified utility to provide more frequent itemsets.
3. The minimum support value determined based on the frequency and utility values also considered the dataset characteristics which include the number of items, number of transactions, and average number of items per transaction.
4. The rules were produced based on the lift ratio value calculated using the utility of the item, thereby allowing the users to see the most relevant rules easily.

It is recommended that future studies focus on pruning and evaluating the adaptive rule model developed in order to determine its performance as well as to implement the model in an actual recommendation system.

## ACKNOWLEDGEMENTS

We would like to thank Institut Teknologi Bandung and Pasim National University for motivational support and direction in writing this article.

### Funding

This research was funded by LPDP (Indonesia Endowment Fund for Education), the Ministry of Finance, and the Republic Indonesia (Grant No. 2020032117861). The funders had no role in study design, data collection and analysis, decision to publish, or preparation of the manuscript.

### Grant Disclosures

The following grant information was disclosed by the authors:
LPDP (Indonesia Endowment Fund for Education), the Ministry of Finance, and the Republic Indonesia: 2020032117861.

### Competing Interests

The authors declare that they have no competing interests.

### Author Contributions

- Erna Hikmawati conceived and designed the experiments, performed the experiments, analyzed the data, performed the computation work, prepared figures and/or tables, authored or reviewed drafts of the article, and approved the final draft.
- Nur Ulfa Maulidevi conceived and designed the experiments, performed the experiments, analyzed the data, prepared figures and/or tables, authored or reviewed drafts of the article, and approved the final draft.
- Kridanto Surendro conceived and designed the experiments, prepared figures and/or tables, authored or reviewed drafts of the article, and approved the final draft.

### Data Availability

The six datasets used in the experiment are available in the Supplemental Files.

### Supplemental Information

Supplemental information for this article can be found online at http://dx.doi.org/10.7717/peerj-cs.1013#supplemental-information.

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
