# Peer review of "Rule-ranking method based on item utility in adaptive rule model"

_PeerJ Computer Science, doi:10.7717/peerj-cs.1013_

## Round 0.1 · original submission · Major Revisions

Your manuscript has not been recommended for publication in its current form. However, we do encourage you to address the concerns and criticisms of the reviewers and resubmit the manuscript once you have updated it accordingly.

·

Basic reporting

Author proposed novel method to find interesting patterns using lift ratio which incorporates both utility and frequency of itemsets in a database. The major highlight of this article is that authors proposed rule ranking method to extract utility based association rules. The article is structured and written well. Though this work addresses the issues of the existing work in utility based data mining, few corrections needs to be done.

1.1. Introduction should clearly addresses the issues of existing works.
1.2. The comprehensive review on utility based data mining should be done. The following recent references should be investigated and cited.
i) https://www.inderscienceonline.com/doi/abs/10.1504/IJITM.2015.066056
ii) https://www.tandfonline.com/doi/full/10.1080/08839514.2014.891839
iii) http://www.cai2.sk/ojs/index.php/cai/article/view/1333
iv) https://ieeexplore.ieee.org/abstract/document/6416812/
v) https://link.springer.com/article/10.1007/s11036-019-01385-6
vi) https://link.springer.com/article/10.1007/s12652-020-02187-5
vii) https://publications.waset.org/9997900/a-distributed-approach-to-extract-high-utility-itemsets-from-xml-data
viii) https://link.springer.com/article/10.1007/s11063-022-10793-x

1.3. Equations should be represented by using equation numbers.
1.4. The flow of algorithm should be clearly explained.
1.5. The algorithm needs to be written clearly with necessary indentations.

Experimental design

2.1 Experimental results needs to be explained elaborated manner.
2.2 Dataset description is not available.
2.3 Experimental results should be supported with necessary graphs by considering various factors such as number of rules generated, support, utility, confidence etc.

Validity of the findings

3.1 The validity and the findings is not clearly explained.
3.2 Statistical analysis can be performed.

Additional comments

The proposed work can be explained with suitable example. Author can use small dataset for this explanation.

·

Basic reporting

No comment

Experimental design

No comment

Validity of the findings

No comment

Additional comments

The authors proposed a method for ranking the rules based on the lift ratio value, which was derived using the item's frequency and utility. I would ask the authors for clarification of some issues before acceptance of this paper. Therefore, I recommend a revision.
• Please, you should add a comparative study. A comparison between your algorithm and others should be added.
• The limitation(s) of the association rules mining methodologies proposed in this work should be extensively discussed.
• When the pseudo-codes of the proposed method are examined, it is seen that concepts such as D, S, X, and U are written in a different language. Please write them all in the same language and give pseudocodes as "appendix".
• Complexity analysis should be done.
• In Figure 1 "eksternal" should be fixed as "external"
• You are using a specific database in Table 1. Please indicate your reasons for choosing these datasets.
• Number of transactions is small in all datasets. A larger data set should be added and analysis should be performed on large data sets.
• I'd like to see a more detailed analysis of the proposed algorithm's scalability. What are the main theoretical and practical benefits of the proposed algorithm? What about memory consumption and problem dimensions (especially big data)?
• "II.2 and II.3" used in " Tables II.2 and II.3 show that the number… " expression should be corrected.
• Is the support value given as a percentage in the test results? Not mentioned in the article?
• The authors mentioned some studies in the literature (such as ELECTRE, ELECTRE II, AHP). They should also compare the proposed method with the existing methods in the literature.
• The properties of the datasets used are already given in Table 1. It is given again in Table 2. If it will be given in Table 1 separately, it should be removed from Table 2. Table 1 is not needed if Table 2 will also be given.

---

## Round 0.2 · accepted · Accept

Thank you for the revision, and for addressing all necessary comments. Your paper has been accepted for publication. The suggested citations do not need to be added.

·

Basic reporting

Authors made significant efforts in improving the manuscript. But, few references i have suggested which are closely associated with their work are not studied and cited. Hence, I suggest authors to study and revise the paper accordingly.

Experimental design

Authors made significant efforts in improving the manuscript. But, few references i have suggested which are closely associated with their work are not studied and cited. Hence, I suggest authors to study and revise the paper accordingly.

Validity of the findings

Authors made significant efforts in improving the manuscript. But, few references i have suggested which are closely associated with their work are not studied and cited. Hence, I suggest authors to study and revise the paper accordingly.

Additional comments

Authors made significant efforts in improving the manuscript. But, few references i have suggested which are closely associated with their work are not studied and cited. Hence, I suggest authors to study and revise the paper accordingly.